# Waning of Humoral Immunity and the Influencing Factors after BNT162b2 Vaccination: A Cohort Study with a Latent Growth Curve Model in Fukushima

**DOI:** 10.3390/vaccines10122007

**Published:** 2022-11-25

**Authors:** Yurie Kobashi, Yoshitake Takebayashi, Makoto Yoshida, Takeshi Kawamura, Yuzo Shimazu, Yudai Kaneko, Yoshitaka Nishikawa, Aya Nakayama, Morihito Takita, Tianchen Zhao, Chika Yamamoto, Naomi Ito, Masaharu Tsubokura

**Affiliations:** 1Department of Radiation Health Management, Fukushima Medical University School of Medicine, Fukushima 960-1247, Japan; 2Department of Internal Medicine, Serireikai Group Hirata Central Hospital, Ishikawa District, Fukushima 963-8202, Japan; 3Department of Health Risk Communication, Fukushima Medical University School of Medicine, Fukushima 960-1295, Japan; 4Faculty of Medicine, Teikyo University School of Medicine, Itabashi-ku, Tokyo 173-8605, Japan; 5Isotope Science Centre, The University of Tokyo, Bunkyo-ku, Tokyo 113-0032, Japan; 6Laboratory for Systems Biology and Medicine, Research Centre for Advanced Science and Technology (RCAST), The University of Tokyo, Meguro-ku, Tokyo 153-8904, Japan; 7Medical & Biological Laboratories Co., Ltd., Minato-ku, Tokyo 105-0012, Japan

**Keywords:** BNT162b2 vaccination, COVID-19, antibody titer kinetics, immunoglobulin G (IgG), humoral immunity

## Abstract

Measuring long-term antibody titer kinetics and subsequent coronavirus disease 2019 (COVID-19) vaccinations are crucial for identifying vulnerable populations. Our aim was to determine the association between long-term antibody kinetics, including peak titers and factors, up to seven months post-second vaccination. A three-time antibody survey was conducted in 2021 among healthcare workers in Japan to investigate the changes in humoral immunity using chemiluminescence immunoassay. The study involved 205 participants who had received the second vaccine dose, completed the three-time survey, and were not infected with SARS-CoV-2. A latent growth curve model was used to identify factors affecting the peak titer and decreasing the antibody slope. Of the eligible participants, the mean titers of immunoglobulin G (IgG) against the spike (S) protein and the neutralizing activity 7 months after the second vaccination decreased to 154.3 (8.8% of the peak titer) and 62.1 AU/mL (9.5% of the peak titer), respectively. The IgG growth model showed that age significantly affected peak titers (*p* < 0.001); however, a significant difference was not found for the decreasing slope. Ultimately, aging adults had significantly low peak antibody titers; however, age was unrelated to the slope of log-transformed IgG against the S protein.

## 1. Introduction

An effective vaccination strategy against coronavirus disease 2019 (COVID-19) is crucial for controlling the pandemic [1,2,3]. Notably, messenger RNA vaccines are promising for infection prevention, owing to their ability to induce cellular and humoral immune responses [4]. However, the titers of immunoglobulin G (IgG) against the spike (S) protein and the neutralizing antibody activity wane over time post-vaccination [5,6]. Specifically, the number of infected people starts to increase a few months after the second dose of the vaccine [7,8,9,10]. The Centers for Disease Control and Prevention (CDC) have recommended a third COVID-19 vaccine dose for immunocompromised individuals [11]. However, because of a shortage of vaccines and increasing vaccine hesitancy, only a few countries have populations that have completed the booster vaccination against COVID-19 [12,13,14]. Thus, it is vital to examine the kinetics of immunity over a long term after the second dose of the vaccine.

Studies have shown the waning of antibody titers after the second dose of the vaccine [5,6,15,16,17], with several recent studies reporting that antibody titers wane substantially among aging adults [5], men [5], immunocompromised hosts [15], individuals who consume alcohol [16], and individuals with smoking habits [17]. Nevertheless, data on long-term humoral immunity among cohorts are lacking. Particularly, factors that affect the waning of long-term humoral immunity in the Japanese population have not been fully elucidated.

The Fukushima Prefecture experienced three disasters in 2011, namely, the radiation disasters as a result of the Fukushima Daiichi Nuclear Power Plant accident, the Great East Japan earthquake, and the subsequent tsunami. Municipalities in disaster-affected areas in Fukushima Prefecture were required to survey radiation exposure and the associated health issues for a decade [18]. During the recent COVID-19 pandemic era, the Fukushima Vaccination Community Survey (FVCS) was conducted in Hirata Village, Soma City, and Minamisoma City, to evaluate antibody titer kinetics after two doses of the BNT162b2 vaccine (Pfizer-BioNTech). Seireikai, a private healthcare group located in the Ken-chu District of Fukushima Prefecture, has been conducting longitudinal antibody surveys for COVID-19 since May 2020 as a strategy for COVID-19 prevention and to determine the infection status among healthcare workers [19,20]. Therefore, this area was deemed suitable for surveying the association between the long-term waning of humoral immunity and the associated factors.

The purpose of this study was to determine the association between long-term antibody kinetics up to 7 months post-second vaccination, including peak titers, and the factors with the latent growth curve model. Our results might help identify vulnerable populations in Japan where these reports are limited. This study was conducted as a part of the FVCS.

## 2. Materials and Methods

### 2.1. Study Participants

This was a historical cohort study. The study participants were recruited from the Seireikai healthcare group, which holds hospitals, nursing home facilities, clinics, among others. Seireikai, located in the depopulated, rural Ken-chu district in Fukushima Prefecture, has been continually conducting antibody testing on healthcare workers since 2020 [19,20] to reveal their infection status and to control infection.

### 2.2. Eligibility Criteria for This Study

The criteria for participants were as follows: (ⅰ) completed a second dose of the BNT162b2 (Pfizer/BioNTech) vaccine, (ⅱ) completed the antibody survey for each term (during the pre-vaccination blood sampling in June (T1), blood sampling in September (T2), and blood sampling in December (T3)), and (ⅲ) not infected with severe acute respiratory syndrome coronavirus 2 (Figure 1). T2 and T3 were supported by the Japan Agency for Medical Research and Development. This study was approved by the ethics committees of Hirata Central Hospital (number 2021-0611-1) and Fukushima Medical University (number 2021-116).

### 2.3. Study Design

A total of 208 participants completed the second-dose vaccination and blood sampling for antibody testing during all terms. First, pre-vaccination blood sampling was performed to measure the antibody titers of immunoglobulin M (IgM) against the nucleocapsid (N) protein immediately before the first dose of BNT162b2 to identify the infection status. The first vaccinations were performed between 19 April and 23 April, and the second between 10 May and 14 May, where the interval between the two doses was 21 days. All participants completed two doses of BNT162b2. The first-term blood sampling was subsequently conducted to measure the titers of IgG against the S protein and the neutralizing activity to identify the increase in antibody titers approximately 21 days after the second vaccination (T1) between 31 May and 6 June. The second-term blood sampling was conducted to measure IgG levels against the S protein, the neutralizing activity, and IgM level against the N protein approximately 129 days after the second vaccination (T2) between 17 September and 22 September. The third-term blood sampling was performed to measure the above levels again approximately 209 days after the second vaccination (T3) between 6 December and 25 December. Of the 208 participants, three individuals who were positive for at least one IgM and IgG against the N protein were excluded from the analysis, to ensure exclusion of past infection. Finally, 205 participants met the eligibility criteria (Figure 1). The data on sex and age were retrieved from a written questionnaire. Written informed consent was obtained from all participants.

The first vaccinations were performed between April 19 and April 23 and the second between 10 May and 14 May, where the interval between the two doses was 21 days. The first-term blood sampling was conducted between 31 May and 6 June (T1). The second term blood sampling was conducted between 17 September and 22 September (T2). The third term blood sampling was performed between 6 December and 25 December (T3).

T1; Time 1. T2; Time 2. T3; Time 3.

### 2.4. Serological Assay

All serological assays were performed using a chemiluminescence immunoassay (CLIA) with the iFlash 3000 (YHLO Biotech, Shenzhen, China) and iFlash-2019-nCoV reagent series (YHLO Biotech). The CLIA is used for the quantitative determination of humoral immunity in human serums with an enzyme and a chemiluminescent. We used the quantitative CLIA because its accuracy is high compare with that of assays using rapid testing kits [19,20]. The trigger was added to the reaction mixture, the resulting chemiluminescent reaction was measured as relative light units, and the results were subsequently determined using a calibration curve [21,22]. The cutoff value of each assay (IgG against the S protein, the neutralizing activity, IgM against the N protein, and IgG against the N protein) was 10 AU/mL following the manufacturer’s official cutoff values. The cut off values were determined using the ROC curve method [21,22]. For neutralizing activity, values over 800 AU/mL were not guaranteed to be accurate according to the manufacturer’s instructions. The testing method was performed according to official guidelines [21,22]. A quality check was conducted daily before starting the measurements. 

### 2.5. Statistical Analysis

We determined the mean titer of IgG against the S protein and the neutralizing activity at each term and for each age group, which are shown in a violin plot. The antibody levels and neutralizing activity were log-transformed for statistical analyses, and the distribution is shown using log-transformed titers. 

We applied the Bayesian latent growth curve model, which aims to examine the average transition by estimating intercepts and slopes (linear and quadratic). We identified the factors affecting the peak titer and the antibody decreasing slope up to 7 months after the second vaccination. Log-transformed antibody titers were used for the analysis. Sex and age (grand–mean centered) were included in the model as time-invariant covariates. All participants were included in the model for IgG against the S protein. To estimate the growth model parameters, Bayesian estimation was used for the analyses (number of iterations = 50,000, burn-in period = first 50% of iterations, number of chains = 2). Priors for Bayesian estimation were non-informative, which was set as the default prior in Mplus (v.8; Muthén & Muthén, Los Angeles, CA, USA). We used potential scale reduction as a measure for appropriate convergence for Bayesian estimation (less than 1.01) and the posterior predictive *p*-value to assess model fit (the closer to 0.50, the better the model). All analyses, except growth curve analysis, were performed using STATA IC (v.15; Lightstone, TX, USA).

## 3. Results

A total of 205 participants were included in the analysis. The median age was 46 (range, min–max: 19–76) years, and 76.1% of the participants were women. Regarding the use of daily medications, 3 used steroids, 13 used non-steroidal anti-inflammatory drugs (NSAIDs), 4 used acetaminophens, 8 used antihistamines, 1 used immunosuppressants, and 1 used biologicals. Some participants had comorbidities—31 had hypertension, 17 had dyslipidemia, 5 had diabetes, 9 had athame, 4 had rheumatism, 15 had allergic disease, 1 had collagen disease, and 1 had immune deficiency. The mean IgG titer against the S protein was 1753.2 arbitrary units per milliliter (AU/mL) at the peak and decreased to 154.3 AU/mL (8.8% of peak titer) 7 months post-second vaccination. The mean neutralizing activity was 657.1 AU/mL at the peak and decreased to 62.1 AU/mL (9.5% of peak titer) 7 months post-second vaccination (Table 1). The kinetics of the mean IgG titers against the S protein and the neutralizing activity of each age group are shown in Table 2. The number of participants in each age group was as follows: 28 in the under-30-years-old, 35 in 30 s, 59 in 40 s, 49 in 50 s, and 34 in the over-59-years-old groups. The mean IgG titers against the S protein and the neutralizing activity were higher among young participants at all three time points. 

The distribution of IgG against the S protein and the neutralizing activity are shown with a violin plot in Figure 2. The IgG levels against the S protein and the neutralizing activity were remarkably decreased 7 months after the second vaccination in all age groups. The mean titers of neutralizing activity at approximately 7 months were 134.9 in the 20’s group and 36.0 in over 60’s group. A comparison between IgG titer and neutralizing activity in each age category is shown in Appendix A. The correlation between IgG titer and neutralizing activity is shown for each age group.

The common log-transformed IgG levels against the S protein and the neutralizing activity are shown in Figure 3. The mean interval was 108 and 80 days between T1–T2 and T2–T3, respectively; thus, the proportion of interval days between the tests was not equal (T1–T2:T2–T3 = 1:0.74). The log-transformed titers also tended to markedly decrease. 

Log-transformed titers of IgG against the S protein were analyzed using a Bayesian latent growth curve model to determine the covariant factors affecting the peak titer and antibody kinetics (Appendix A). The quadratic growth curve model was convergent (potential scale reduction was less than 1.01), and well-fitted to the data (post-predictive *p* = 0.464). The mean intercept, slope, and quadratic were 3.200, −1.039, and 0.231 respectively (Table 3). The covariance between the peak titer (intercept) and linear slopes was significantly positively related; the higher the peak titer, the less steep the linear slope of log-transformed IgG against the S protein. Age was significantly positively related to the peak titer; however, no significant relationship was observed with the slopes. In contrast, sex was not significantly related to the peak titer, or the linear or quadratic slopes.

## 4. Discussion

Here, the peak antibody titers were affected by age, with aging adults showing a significantly lower peak antibody titer. Similarly, previous studies have shown that age is associated with COVID-19 morbidity and mortality [23,24]. The peak antibody titers after vaccination were lower among aging adults, which is a noteworthy factor that should be utilized when considering vaccine measures.

The lower humoral immunity among aging adults may be caused by the lower peak of antibody titers rather than the faster waning of antibody titers. Here, age significantly affected the peak log-transformed antibody titers; however, no significant difference was observed in the slope of the waning antibody titers. An approach to obtain higher antibody titers after vaccination among aging adults is of high importance. 

In this study, the antibody titers after the second vaccine dose were decreased among all age groups. The neutralizing activity was 134.9 AU/mL in the 20’s group 7 months after the second dose of vaccine, which might be sufficient to prevent infection and severe symptoms. Previous studies have shown that breakthrough infections can spread even among young people several months after the second vaccination [7,8,25]. A more comprehensive evaluation of the immune response against COVID-19 after the second vaccination is required.

Some limitations should be considered when interpreting the results of this study. First, the number of participants included was limited, thus the results might be affected. Second, neutralizing activity titers over 800 AU/mL were not guaranteed to be accurate, and we found a loss in correlation between neutralizing activity over 500 AU/mL and IgG levels against the S protein. Additionally, in our data, neutralizing activity over 500 AU/mL were not linearly correlated with IgG against S protein titers. Third, antibody titers were log-transformed for latent growth curve model analyses. We should be cautious when considering the application of our results to the antibody titers themselves. Fourth, the latent growth model used here did not consider essential information such as previous medical history or medications. Fifth, a part of the data of the present study was reported previously [26,27,28]; however, the participants were not overlapped, and the analysis method and purpose were different from those in the previous studies. Finally, the sample cohort consisted mainly of healthy healthcare workers, which does not necessarily reflect the general population. Despite these limitations, this study is the first to investigate the factors associated with antibody titer kinetics up to 7 months after the second-dose vaccination in the Japanese population.

## 5. Conclusions

Here, the log-transformed peak antibody titers in aging adults were significantly lower than those in younger adults; however, age was unrelated to the decreasing slope of the log-transformed IgG. The lower humoral immunity among aging adults might be caused by the lower peak of antibody titers rather than the faster waning of antibody titers. Further research is required to understand individual differences in antibody titer waning.

## Figures and Tables

**Figure 1 vaccines-10-02007-f001:**
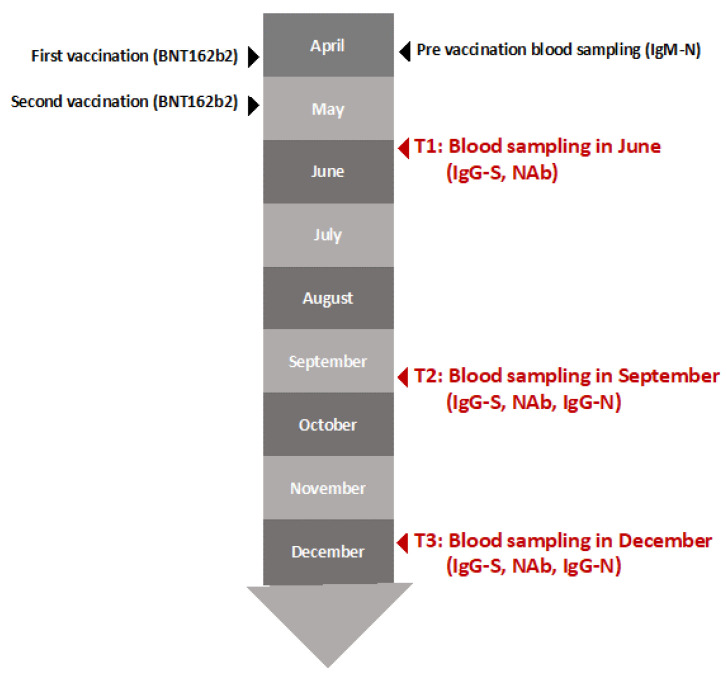
Time schedule for blood sampling from the participants.

**Figure 2 vaccines-10-02007-f002:**
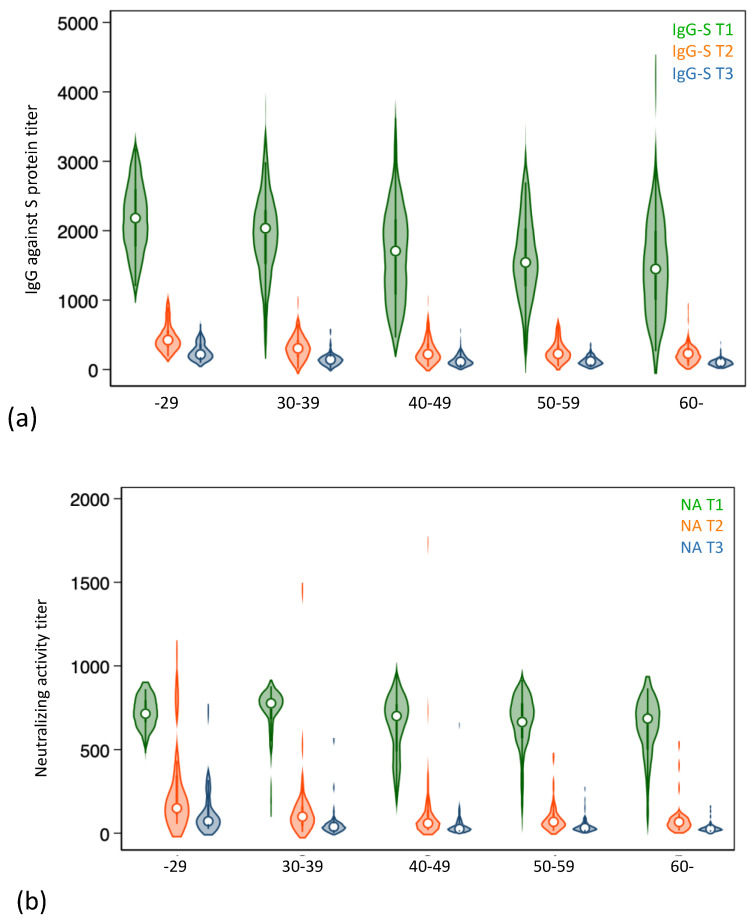
Distribution of antibody titers. (**a**) Violin plot showing the distribution of IgG against the S protein. (**b**) Violin plot showing the distribution of neutralizing activity.

**Figure 3 vaccines-10-02007-f003:**
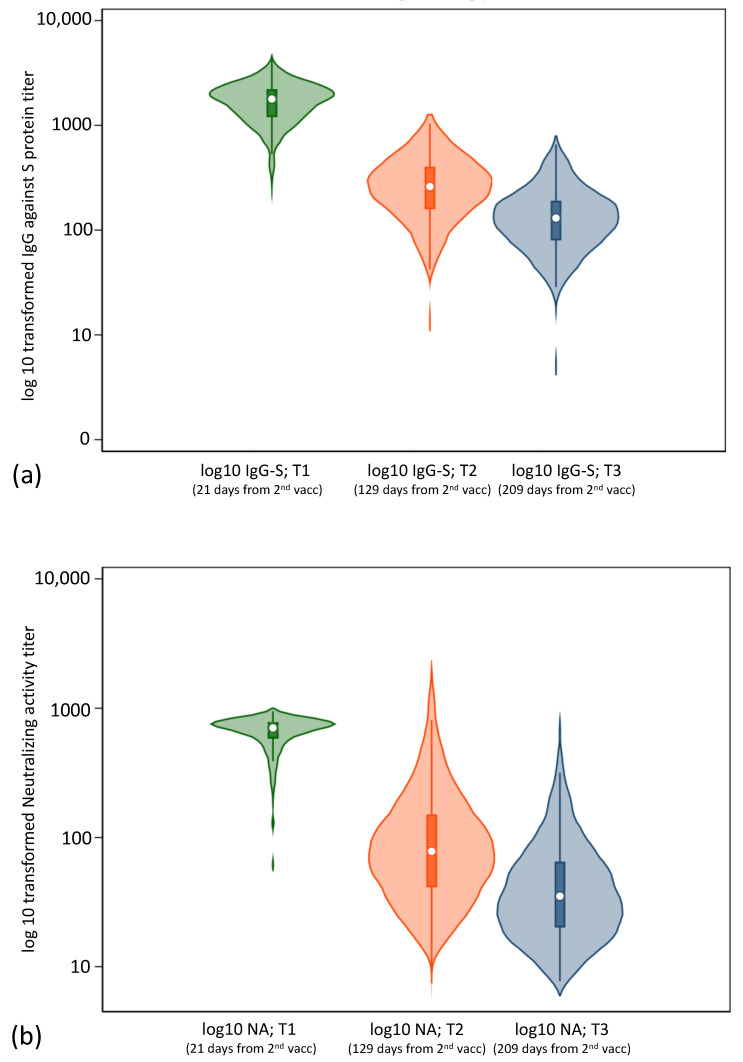
Common log-transformed antibody titers. (**a**) Violin plot showing the common log-transformed IgG against the S protein. (**b**) Violin plot showing the common log-transformed neutralizing activity.

**Table 1 vaccines-10-02007-t001:** Participant characteristics (N = 205).

	Mean	SD
Age	46.0	12.9
Sex male (*n* (%))	49	23.9
June		
Day from the second vaccination	21.1	0.6
Neutralizing activity	657.1	181.3
IgG against S protein	1753.2	712.7
September		
Day from the second vaccination	129.0	2.5
Neutralizing activity	140.4	211.0
IgG against S protein	306.1	199.8
December		
Day from the second vaccination	209.3	2.8
Neutralizing activity	62.1	91.6
IgG against S protein	154.3	109.0

**Table 2 vaccines-10-02007-t002:** Mean of antibody titer by age groups (N = 205).

Age Group (Years)	IgG against S Protein (Mean [SD])			Neutralizing Activity (Mean [SD])		
	June	September	December	June	September	December
29≤	2161.2 (526.2)	479.2 (215.9)	270.0 (134.6)	720.7 (97.6)	273.8 (271.9)	134.9 (148.1)
30–39	1945.0 (704.1)	326.4 (196.2)	152.0 (104.4)	718.3 (160.6)	157.9 (248.3)	63.4 (98.9)
40–49	1692.3 (711.3)	276.3 (193.0)	141.1 (101.0)	639.5 (181.8)	129.7 (244.0)	53.5 (87.5)
50–59	1611.5 (644.9)	270.3 (162.5)	134.5 (82.1)	633.5 (196.9)	99.9 (98.5)	47.8 (50.4)
≥60	1529.6 (800.4)	245.7 (178.8)	112.9 (74.9)	606.4 (207.2)	89.1 (108.1)	36.0 (33.5)

The number of participants in each age group was as follows: 28 in the under 30-years-old, 35 in 30 s, 59 in 40 s, 49 in 50 s, and 34 in over 59-years-old groups.

**Table 3 vaccines-10-02007-t003:** Estimated parameters from the latent growth curve models.

Predictor	IgG(S)		
	Intercept	Slope	Quadratic
Sex	−0.013 (−0.081, D80.055)	−0.071 (−0.167, 0.026) *	0.029 (−0.012, 0.070) *
Age	−0.005 (−0.007, −0.003) ***	−0.002 (−0.005, 0.001)	0.000(−0.002, 0.002)
Mean	3.200	−1.039	0.231
Correlation (I and S)	0.436 (0.161, 0.890) ***		
Correlation (I and Q)	−0.605 (−0.966, −0.311) ***		
Correlation (S and Q)	−0.937 (−0.979, −0.883) ***		

Note: *** *p* < 0.001, * *p* < 0.1.

## Data Availability

The data that support the findings of this study are available from the Fukushima Medical University School of Medicine; however, restrictions apply to the accessibility of these data, which were used under license for this study, as they are not publicly available. However, data are available from the authors on reasonable request and with permission from Fukushima Medical University School of Medicine.

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
