# Peer review of "Waning of Humoral Immunity and the Influencing Factors after BNT162b2 Vaccination: A Cohort Study with a Latent Growth Curve Model in Fukushima"

_vaccines, 2022, doi:10.3390/vaccines10122007_

Round 1

Reviewer 1 Report

I read with great interest this historical cohort study on the factors impacting waning humoral immunity following vaccination with the Pfizer-BioNTech BNT162b2 primary series. Data on the long-term antibody titer kinetics associated with the second Pfizer COVID-19 vaccine dose is critical to improving vaccine efficacy and informing the need for booster immunizations. Identifying factors associated with waning humoral immunity against COVID-19 will be important in targeting vulnerable populations for additional prevention measures. A larger sample size and more diverse study participant population are needed for future studies.  My specific comments are provided below:

(1) Table 2 (Mean of antibody titer by age groups):  For the data category headings, the authors should include "Age Group".  In addition, age group "-29" should be presented as ≤ 29; age group "60-" should be presented as ≥ 60. Please make these corrections in Table 2 for clarity.

(2) What was n for each age group?  Please include this information in the text or Table 2.   

Author Response

Reviewer 1

I read with great interest this historical cohort study on the factors impacting waning humoral immunity following vaccination with the Pfizer-BioNTech BNT162b2 primary series. Data on the long-term antibody titer kinetics associated with the second Pfizer COVID-19 vaccine dose is critical to improving vaccine efficacy and informing the need for booster immunizations. Identifying factors associated with waning humoral immunity against COVID-19 will be important in targeting vulnerable populations for additional prevention measures. A larger sample size and more diverse study participant population are needed for future studies.  My specific comments are provided below:

Response:

We thank you for reviewing our manuscript and providing constructive suggestions and comments. We have revised the manuscript in accordance with the comments. The following are our responses to the comments.

(1) Table 2 (Mean of antibody titer by age groups):  For the data category headings, the authors should include "Age Group".  In addition, age group "-29" should be presented as ≤ 29; age group "60-" should be presented as ≥ 60. Please make these corrections in Table 2 for clarity.

Response: We thank you for your comment. We have revised Table 2 accordingly.

(2) What was n for each age group?  Please include this information in the text or Table 2.   

Response: We thank you for your comment. We have inserted the following sentence in the footnote of Table 2 and the Results.

Table 2 footnote:

“The number of participants in each age group was as follows: 28 in the under 30-years-old, 35 in 30s, 59 in 40s, 49 in 50s, and 34 in over 59-years-old groups.”

Results:

“The number of participants in each age group was as follows: 28 in the under 30-years-old, 35 in 30s, 59 in 40s, 49 in 50s, and 34 in over 59-years-old groups.”

Reviewer 2 Report

the main strength of the current study is the availability of sera of HCWs without COVID-19 prior to the implementation of vaccination against the disease. Moreover, detection of N IgM/IgG and exclusion of individuals with positive results is a guarantee that only non-infected people had participated in the study. This is a main point to be highlighted, however, low ( or lower compared by age group) antibodies' titer cannot prelude sensitivity to infeciton in HCWs taking into account the potential bias of highly applied protective measures compared to other population groups.

Minor comments: 1) you should report that there is no ovelapping of the population study with previous pubslihed data by the same group 2)  A more extensive description of the CLIA of the Yholo company 3) a comment on differences in  the sensitivity of various  commercial kits as already mentioned by the WHO- the kit that was used is not reported ( doi:10.1016/j.intimp.2021.108095), however, there is a publication ( ref  20) by the same  scientific group and another recent one (doi:10.3389/fmicb.2022.876227.  4) a report of comorbiditiesor therapies  relevant to immunocompromized  host among participants-if any exist especially in older age-should be reported 5) a figure comparing IgG and NA at each age category could be added 6) results could be also depicted by ROC-the cutt of value of detection should be also added at Methods. 7) Introduction: the aim of the study is twice reported, just keep it once

Author Response

Reviewer 2
the main strength of the current study is the availability of sera of HCWs without COVID-19 prior to the implementation of vaccination against the disease. Moreover, detection of N IgM/IgG and exclusion of individuals with positive results is a guarantee that only non-infected people had participated in the study. This is a main point to be highlighted, however, low ( or lower compared by age group) antibodies' titer cannot prelude sensitivity to infeciton in HCWs taking into account the potential bias of highly applied protective measures compared to other population groups.

Response:

We thank you for reviewing our manuscript and providing constructive suggestions and comments. We have revised the manuscript in accordance with the comments. The following are our responses to the comments.

1) you should report that there is no ovelapping of the population study with previous pubslihed data by the same group

Response: We thank you for your comment. A part of the data was reported previously; however, the published results have not been presented elsewhere. We have mentioned this in the limitations section as follows.

“Fifth, a part of the data of the present study was reported previously [26­–28]; however, the participants were not overlapped and the analysis method and purpose were different from those in the previous studies.”

2)  A more extensive description of the CLIA of the Yhlo company

Response: We thank you for your comment. We have inserted the following sentence in the serological sassy section.

“The CLIA is used for the quantitative determination of humoral immunity in human serums with an enzyme and a chemiluminescent. We used the quantitative CLIA because its accuracy is high compare with that of assays using rapid testing kits. The trigger was added to the reaction mixture, the resulting chemiluminescent reaction was measured as relative light units, and the results were subsequently determined using a calibration curve.”

3) a comment on differences in the sensitivity of various  commercial kits as already mentioned by the WHO- the kit that was used is not reported (doi:10.1016/j.intimp.2021.108095), however, there is a publication ( ref  20) by the same  scientific group and another recent one (doi:10.3389/fmicb.2022.876227.  

Response: We thank you for your comment. We used the quantitative CLIA instead of an assay using a kit because the accuracy of CLIA might be high compared with that of assays using rapid testing kits. We have inserted the following sentence in the methods section to clarify the above.

“We used the quantitative CLIA because its accuracy is high compare with that of assays using rapid testing kits.”

4) a report of comorbidities or therapies relevant to immunocompromized host among participants-if any exist especially in older age-should be reported

Response: We thank you for your comment. We have inserted the information of comorbidities and medications used among participants in the results section.

“Regarding the use of daily medications, 3 used steroids, 13 used NSAIDs, 4 used acetaminophen, 8 used antihistamines, 1 used immunosuppressants, and 1 used biologicals. Besides, some participants had comorbidities—31 had hypertension, 17 had dyslipidemia, 5 had diabetes, 9 had athame, 4 had rheumatism, 15 had allergic disease, 1 had collagen disease, and 1 had immune deficiency.”

5) a figure comparing IgG and NA at each age category could be added

Response: We thank you for your comment. We have inserted a supplementary figure and the following sentence in the results section.

“A comparison between IgG titer and neutralizing activity in each age category is shown in Supplementary Figure 1. The correlation between IgG titer and neutralizing activity is shown for each age group.”

6) results could be also depicted by ROC-the cut of value of detection should be also added at Methods.

Response: We thank you for your comment. The cut off values were decided using the ROC curve; however, the figures were not shown in the manufacturer’s guidelines. We have inserted the following sentence in the serological assay section.

“The cut off values were determined using the ROC curve method.”

7) Introduction: the aim of the study is twice reported, just keep it once

Response: We thank you for your comment. We have deleted the duplicated sentence in the introduction section.